# Photo-Excited Metasurface for Tunable Terahertz Reflective Circular Polarization Conversion and Anomalous Beam Deflection at Two Frequencies Independently

**DOI:** 10.3390/nano13121846

**Published:** 2023-06-12

**Authors:** Zhixiang Xu, Cheng Ni, Yongzhi Cheng, Linhui Dong, Ling Wu

**Affiliations:** 1School of Information Science and Engineering, Wuhan University of Science and Technology, Wuhan 430081, China; 15871952323@163.com (Z.X.); 15871366321@163.com (C.N.); d15984371276@163.com (L.D.); 2Engineering Research Center for Metallurgical Automation and Detecting Technology Ministry of Education, Wuhan University of Science and Technology, Wuhan 430081, China; 3Hubei Longzhong Laboratory, Xiangyang 441000, China; 4School of Physics and Electronic Information Engineering, Hubei Engineering University, Xiaogan 432000, China; ruochen143abc@163.com

**Keywords:** metasurface, circular polarization conversion, photoconductive silicon, terahertz, tunable reflective deflector

## Abstract

In this paper, a photo-excited metasurface (MS) based on hybrid patterned photoconductive silicon (Si) structures was proposed in the terahertz (THz) region, which can realize the tunable reflective circular polarization (CP) conversion and beam deflection effect at two frequencies independently. The unit cell of the proposed MS consists of a metal circular-ring (CR), Si ellipse-shaped-patch (ESP) and circular-double-split-ring (CDSR) structure, a middle dielectric substrate, and a bottom metal ground plane. By altering the external infrared-beam pumping power, it is possible to modify the electric conductivity of both the Si ESP and CDSR components. By varying the conductivity of the Si array in this manner, the proposed MS can achieve a reflective CP conversion efficiency that ranges from 0% to 96.6% at a lower frequency of 0.65 THz, and from 0% to 89.3% at a higher frequency of 1.37 THz. Furthermore, the corresponding modulation depth of this MS is as high as 96.6% and 89.3% at two distinct and independent frequencies, respectively. Moreover, at the lower and higher frequencies, the 2π phase shift can also be achieved by respectively rotating the oriented angle (α_i_) of the Si ESP and CDSR structures. Finally, an MS supercell is constructed for the reflective CP beam deflection, and the efficiency is dynamically tuned from 0% to 99% at the two independent frequencies. Due to its excellent photo-excited response, the proposed MS may find potential applications in active functional THz wavefront devices, such as modulators, switches, and deflectors.

## 1. Introduction

Terahertz (THz) radiation pertains to electromagnetic (EM) waves within the frequency range of 0.1 THz to 10 THz. In comparison to X-rays and microwaves, THz radiation exhibits a smaller single photon energy and a more diverse spectral range. Consequently, the technologies and applications derived from THz radiation have tremendous potential for development in several fields [1], such as biomedicine [2,3,4], astronomy [5], and next-generation wireless communications [6]. Nevertheless, progress in the development of THz technologies and applications has been limited due to the difficulty of naturally occurring matter producing an efficient EM response to THz waves [7,8]. In the past few years, the THz frequency range was once known as the “THz Gap” [9]. However, with the advent of EM metamaterials (MMs) [10,11,12,13], the research in THz technologies and applications has gained significant traction. MMs are man-made materials that consist of artificially designed microcellular structures arranged in specific arrangements. Despite being composed of natural materials at their core, MMs can achieve a series of EM wave modulation phenomena that the scope of natural materials cannot achieve, such as negative refraction [14,15], imaging [16,17], transformation optics [18], perfect absorption [19], and deflection [20,21], among others.

To overcome the limitations of high loss, large size, and complex fabrication techniques associated with three-dimensional (3D) MM themselves [22], researchers have developed two-dimensional (2D) ultrathin arrays, called metasurfaces (MSs) [23,24,25]. Compared to MMs, MSs offer the advantage of being lighter, thinner, and easier to fabricate, while also providing more flexible modulation of THz waves. Thus, various THz devices with single or composite functions, such as abnormal refraction/reflection, perfect absorption, beam splitters, orbital angular momentum, polarization conversion, beam deflection, and so on, have been proposed intensively [26,27,28,29,30,31,32,33,34,35,36,37,38,39,40,41]. Among these functions, polarization conversion and beam deflection are some of the most fundamental functionalities in photoelectric information processing systems. In recent years, a large number of MSs from microwave to even visible frequencies have been proposed and investigated intensively, which can be functioned as various polarization converters and beam deflectors [33,34,35,36,42,43]. For example, Zhao et al., proposed a transmission-type MS based on bilayer metallic patches, which functioned as a high-efficiency THz beam deflector [33]. Cheng et al., designed a tri-band high-efficiency circular-polarization (CP) converter based on a tri-layer split-ring structure, which can convert the incident CP wave to its orthogonal component after transmission [34]. Liu et al., demonstrated a novel design of a deeply subwavelength structure, which was constructed as a THz reflective beam deflector [35]. Jia et al., proposed an achromatic dielectric MS, which demonstrated an achromatic feasibility of the beam deflector from 0.6 to 1.2 THz [36]. Although the THz MSs can achieve the polarization conversion or beam deflection function, most functionalities are fixed and unchangeable once they are designed and fabricated, thus restricting their scope of application. As the application blooms increasingly, developing tunable MSs is necessary and highly desirable [44,45,46,47,48,49]. Ha et al., proposed a new strategy for on-chip planar compact MS made of gold and phase-change material GST, which can achieve a continuous beam-steering at 10.6 μm [46]. Kim et al., proposed an innovative approach for designing a nanophotonic beam deflector with superior performance characteristics, including broadband operation, extensive active tunability, and high signal efficiency, all achieved simultaneously [47]. Yu et al., proposed a novel and reconfigurable MS capable of dynamically switching the deflection angles in the opposite direction, while also exhibiting an impressive relative bandwidth of up to 47.6% for achromatic reflection in the microwave region [48]. Recently, photoconductive Si has increasingly received great attention due to its active tunable behavior of the electric conductivity induced by external optical pumping power [49]. The photoconductive Si can be used to construct the various MS devices with tunable functions in the THz region [49,50,51,52,53,54,55,56,57,58,59], such as absorbers, switchers, and polarization converters. The patterned photoconductive Si is usually embedded in or combined with the metal resonator to form a hybrid structure for dynamically manipulating the THz wave, which can be prepared and fabricated easily using standard lithography and E-beam evaporation technology [49,55,56]. The electric response of the patterned photoconductive Si is determined by the infrared pumping power. However, photoconductive Si-based tunable MSs with functionalities such as polarization conversion and beam deflection have not been studied yet.

In this work, we have designed and demonstrated a photo-excited THz tunable MS based on hybrid structures of photoconductive Si ellipse-shaped patch (ESP) and circular double split ring (CDSR) and metal circular ring (CR), which can be used to construct a tunable reflective CP converter and a beam deflector, respectively. Since the conductivity of the patterned Si structures can be adjusted by changing the optical-pump power, the proposed MSs can realize a continuous modulation of the efficiency of the CP conversion and deflection at two independent THz frequency ranges. Firstly, the reflection CP conversion property of the proposed MS was demonstrated numerically under the upper optical-pump power. The physics origin of the proposed MS structures was illustrated by analyzing the electric field and surface current distributions on the unit cell at two different frequencies. Then, by respectively rotating the azimuth rotation angles (*α*_1_ and *α*_2_) of the Si CDSR and ESP structures, 0–2π phase shift of the reflection orthogonal CP waves could be obtained around 0.65 THz and 1.37 THz, respectively. Thirdly, a tunable MS reflective-beam deflector with a linear phase gradient was proposed and demonstrated numerically. This optically tunable reflective-mode MS may find more exciting applications, such as polarization converters, modulators, and beam deflectors.

## 2. Structure Design and Simulations

Figure 1 illustrates the scheme of the proposed photo-excited THz tunable MS. As shown in Figure 1b,c, the unit cell of the proposed photo-excited MS is composed of hybrid structures of Si ESP and CDSR and metal CR, a middle dielectric substrate, and a metal ground plane. The top and bottom layers of the proposed MS structure can form a Fabry–Pérot-like resonance cavity, resulting in a highly efficient polarization conversion through the multiple interference effect [43,53]. In addition, the symmetry axis of ESP and CDSR structures of the proposed MS is rotated *α_i_* (*i* = 1, 2) along the *x*(*y*)-axis direction, indicating a typical anisotropy in the *x-y* plane. This anisotropic design is also very key to achieving reflective polarization conversion [36,60,61,62,63]. Note that the two distinct resonators (Si ESP and CDSR structure) are responsible for manipulating the THz wave at two corresponding frequencies independently. The sequence arrangement of discrete ESP and CDSR structures will inevitably lead to cross-coupling effects, which will cause some discrepancies between the designed phase and the practical arranged phase, thus finally resulting in a reduction of MS device efficiency [31,32]. In our design, the metal CR structure is added and used to reduce the cross-talk and cross-coupling effect between the ESP and CDSR structures. By constantly changing the radius of the metal CR, we ultimately controlled the impact to a low degree. Thus, it can be expected that the designed MS can independently control the reflective beams at two different THz frequency regions.

The electric conductivity of the photoconductive Si is usually proportional to the pumping power of the incident infrared light [49,50]. For example, the conductivity of the photoconductive Si is only about 1 S/m without infrared pumping (power is zero), while the one is about 1.5 × 10^5^ S/m under the upper-limiting infrared pumping (central wavelength of 800 nm and power is about 405 mJ/cm^2^) [49,50,55]. As shown in Figure 1a, an excess carrier density will be generated as long as the pumping energy exceeds the band-gap energy of the photoconductive Si when an infrared beam is illuminated on the designed MS [49,50]. In this case, the hybrid structures of the photoconductive Si array are in the metal state. While without infrared illumination, the hybrid structures of the photoconductive Si are in the insulation state. Note that the entire area of the designed MS is fully covered by the infrared beam (see Figure 1a).

As is well known, the geometric phase, also named Pancharatnam–Berry (PB) phase, has the obvious advantage of being nondispersive of the MS unit cell. The spatial asymmetry of the MS unit cell will lead to the anisotropy of the polarized wave. The oriented angles (*α*_1_ and *α*_2_) of the designed MS exhibit spatially dependent optical axis distribution, which will bring the additional phase difference *φ* to the two orthogonal CP components. According to the PB principle, the relationship between the oriented angles (*α*_1_ and *α*_2_) of the structural photoconductive Si and the additional phase *φ* can be expressed by *φ_i_* = 2*σα_i_*, where *σ* = ±1 corresponds to the helicity direction for the CP wave [26], e.g., left-handed or right-handed CP (“+” RCP, or “−” LCP). Consequently, the continuous 2π phase shift can be easily achieved by rotating the *α_i_* (*i* = 1, 2) instead of changing the structure parameter of the MS unit cell.

To demonstrate the functions of the designed MS, a full wave simulation was carried out using the frequency solver by the CST Microwave Studio based on the finite element method (FEM). In the simulation, the Floquet ports for the incident THz CP plane waves along the *z*-axis direction are assigned to the unit cell of the proposed MS structure. In addition, the distance between the wave ports and the unit-cell structure is 300 μm in order to only get enough propagating information of the reflected THz wave. In addition, the minimized mesh size is set as 0.22 μm, which is much smaller than the unit-cell size and the operating wavelength. During the optimization process, the main objective is to get the maximized magnitude of the reflected orthogonal CP wave at two distinct THz frequency ranges when the designed MS is under the upper infrared pumping power (*σ*_si_ = 1.5 × 10^5^ S/m). The hybrid structure of photoconductive Si was modeled as a dielectric with a relative permittivity of *ε*_Si_ = 11.7, while the electrical conductivity *σ*_Si_ is dependent on external infrared beam power in our interested THz frequency range [49,50]. The metal CR structure and ground plane of the MS are made of a copper film with conductivity σ = 5.8 × 10^7^ S/m. The low-loss polyimide with a permittivity of 2.35 × (1 + *i* × 0.0027) was chosen as the middle dielectric substrate layer. After simulation optimization, the final geometrical parameters of the unit cell are given as *p_x_ = p_y_* = 100 μm, *a* = 26 μm, *b* = 8 μm, *r* = 40 μm, *w* = 5 μm, *g* = 20 μm, *t_m_* = 3 μm, *r*_0_ = 33.5 μm, *w*_0_ = 1.5 μm, and *t_s_* = 35 μm. In addition, the orientations of the outer CDSR and inner ESP are denoted as *α*_1_ and *α*_2_ with respect to the *x/*(*y*)-axis, respectively.

## 3. Results and Discussion

Figure 2 presents the reflection amplitude and polarization conversion ratio (PCR) for the structure under different infrared pump powers (the corresponding conductivity of photoconductive Si is from *σ*_si_ = 1 S/m to *σ*_si_ = 1.5 × 10^5^ S/m) at the lower and higher frequency ranges. Clearly when increasing the conductivity of photoconductive Si from 1 S/m to 1.5 × 10^5^ S/m, the amplitude and PCR of the reflected orthogonal CP wave for the normal incident RCP(LCP) wave will increase continuously at 0.4–0.9 THz and 1.2–1.5 THz, respectively. Here, we define the polarization conversion capability of the device as PCR:(1)$$\{\begin{array}{c} {\mathrm{PCR}}_{+}=\frac{{\left|r_{-+}\right|}^{2}}{{\left|r_{-+}\right|}^{2}+{\left|r_{++}\right|}^{2}}\\ {\mathrm{PCR}}_{-}=\frac{{\left|r_{+-}\right|}^{2}}{{\left|r_{+-}\right|}^{2}+{\left|r_{--}\right|}^{2}} \end{array}$$
where the first subscript indicates the polarization state of the reflected wave, and the second subscript indicates the polarization state of the incident wave. For example, *r_−+_* and *r_+−_* represent the orthogonal CP reflection coefficients, and the *r_++_* and *r_−−_* denote the copolarization CP reflection coefficients.

As shown in Figure 2a,b, when *σ*_si_ = 1 S/m, the reflection amplitude of the orthogonal CP wave for the proposed MS structure is only about 0.02 and 0.03 on average, and the corresponding PCR is near zero across the whole lower (0.4–0.9 THz) and higher (1.2–1.5 THz) frequency ranges (see Figure 2c,d). It reveals that nearly no reflective CP conversion occurs without infrared beam illumination, and the designed structure is in an “OFF” state in this case. In the absence of infrared beam illumination (*σ*_si_ = 1 S/m), the outer CDSR and inner ESP photoconductive Si are in an isolation state, which can not effectively respond to the incident CP wave. When the structure is under the upper infrared pumping power (*σ*_si_ = 1.5 × 10^5^ S/m), the reflection amplitude is up to the maximal values of 0.816 and 0.709, and the PCR is up to 96.6% and 89.3% at 0.65 THz and 1.37 THz, respectively, revealing the “ON” state in this case. It means that the reflective CP conversion efficiency from 0 to 96.6% and 0 to 89.3% can be adjusted dynamically through varying the external infrared pumping power of the designed MS at the above two distinct frequencies, respectively. To further illustrate the tunability of the proposed MS structure for the CP conversion, the modulation depth of the CP conversion is defined as *d* = PCR_max_ − PCR_min_, where the PCR_max_ and PCR_min_ are the CP conversion efficiency when the proposed MS structure is under the upper limiting infrared pumping power (*σ*_si_ = 1.5 × 10^5^ S/m) and without infrared pumping power (*σ*_si_ = 1 S/m). Thus, the corresponding modulation depth of the CP conversion of the proposed structure is near 96.6% and 89.3% at the lower and higher frequencies, respectively. Therefore, the reflective CP conversion efficiency of the proposed MS can be tuned actively by changing the conductivity of the photoconductive Si through changing the infrared pumping power at two independent frequency ranges.

To illustrate the physics mechanism of the proposed MS for the CP conversion at two independent frequencies, we presented the induced z-component of the electric field (|*E_z_*|) and surface current distributions on the unit-cell structure for the normal incident LCP wave under the upper infrared pumping power (*σ*_si_ = 1.5 × 10^5^ S/m) at 0.65 THz and 1.37 THz, respectively.

Since both the outer CDSR and inner ESP structures are assumed as the evolved metal cut-wire structure resonators, thus, a similar electric response can be observed at the two distinct frequencies. As shown in Figure 3a,b, the induced z-component of the electric field (|*E_z_*|) is mainly focused and distributed on the outer Si CDSR structure at the 0.65 THz, while the one is on the inner Si ESP structure at 1.37 THz. It means that the outer CDSR structure corresponds to the resonance response of the lower frequency, while the inner ESP structure to the one of the higher frequency. The electric fields of both the incident and the reflected CP waves are decomposed into two orthogonal components, i.e., *u*-component and *v*-component, respectively. The eigenmode resonances on the Si ESP and CDSR structures can be excited with electric components along the *u-* and *v*-axis directions, respectively. The excited dipole of the Si ESP and CDSR partly couples to the dipole mode on the interface of the top layer and air, leading to cross-polarization conversion after reflection. Obviously, the |*E*_z_| distributions on the Si ESP and CDSR structures reveal effective excitation of the surface plasmon polaritons (SPPs) at resonances. As shown in Figure 3c,d, at the lower frequency of 0.65 THz, the induced surface current is mainly distributed on the upper and down edges of the outer CDSR structure, while the one is on the left-up and right-down edges of the part inner ESP structure at the higher frequency of 1.37 THz, which are both consistent well with the induced electric-field distributions along the *v*-axis direction, revealing strong electric dipolar resonance responses of the unit-cell structure for the normal incident LCP wave. Note that nearly no surface current distributions can be observed on the middle metal CR structure at the above two distinct frequencies. These electric-field and surface-current distributions on the unit cell reveal that the reflective CP conversion of the designed structure mainly originates from the fundamental dipolar resonance and structure anisotropy, showing that the respective control at two independent frequency regions is available.

To achieve the efficient reflective beam deflection effect for the CP wave, the full 2π phase coverage should be satisfied at the two distinct frequency ranges by rotating the *α_i_* (*i* = 1, 2) based on the PB phase principle instead of changing the structure parameter of the MS. Figure 4 presents the reflection amplitude and phase of the orthogonal CP wave of the proposed structure with different rotation angles (*α*_1_ and *α*_2_) under the upper infrared pump power (*σ*_si_ = 1.5 × 10^5^ S/m). As shown in Figure 4a,c, at the lower frequency range of 0.5–0.9 THz, the designed MS exhibits a linear gradient phase of the reflected orthogonal CP wave when rotating the *α*_1_ of the outer CDSR structure by a step of 22.5° from 0° to 157.5°, while the corresponding amplitude is nearly unchanged. As shown in Figure 4b,d, at the higher frequency range of 1.2–1.5 THz, similar results for the reflection amplitude and phase also can be obtained by rotating the *α*_2_ of the inner ESP structure by a step of 22.5° from 0° to 157.5°. Obviously, the reflection amplitude of the orthogonal CP wave through the MS is very high and the corresponding phase is varied linearly with a certain interval by changing the *α*_1_ and *α*_2_. It means that the orthogonal reflection phase can realize the full coverage from 0 to 2π by respectively changing the *α*_1_ and *α*_2_ from 0° to 180° at the two independent frequency regions.

In the flowing section, we will study the THz beam deflection effect for the proposed MS based on the PB phase principle under infrared illumination with different pumping power. Here, as shown in Figure 5a, a supercell comprised of eight unit cells with different *α*_1_ and *α*_2_ has been designed to guarantee that the linear phase gradient from 0 to 2π with high efficiency can be achieved at two different frequency ranges. As shown in Figure 5b,c, when the designed supercell is illuminated by the infrared with upper pumping power (*σ*_si_ = 1.5 × 10^5^ S/m) at *f*_L/_*f*_H_ = 0.65/1.37 THz, the rotating angles *α*_1_/*α*_2_ of eight unit cells are 0°/−0.9°, 45°/45°, 90°/90°, 135°/135°, 178°/179°, −134°/−136°, −89°/−89°, and −46°/−44°, respectively. Obviously, the designed MS supercell exhibits a discontinued phase shift with nearly a step of 45° at 0.65 THz and 1.37 THz, respectively. In addition, as illustrated in Figure 5b,c, all reflection amplitudes for the selected eight unit cells of the designed supercell are about 0.8 and 0.65 on average at 0.65 THz and 1.37 THz, respectively, which is favorable to achieve the highly efficient beam deflection effect. By introducing a linear phase gradient, it is expected that the designed MS supercell can realize a THz beam deflection effect with continuous tunable efficiency under different infrared pumping power.

Based on the generalized Snell’s law, the beam deflection effect can be illustrated through the following equation [9,10]:(2)$$n_{r}\sin\theta_{r}-n_{i}\sin\theta_{i}=\frac{\lambda }{2\pi }\frac{d\Phi (x)}{dx}$$
where the incident angle is denoted as *θ_i_*, while the reflection angle is represented by *θ_r_*. The refraction indices of the medium in the incidence and reflection spaces are denoted as *n_i_* and *n_r_*, respectively. The *λ* refers to the operation wavelength in free space, and *d*Φ(*x*)*/dx* along the *x* direction signifies the phase gradient along the designed supercell interface. Using Equation (2), it can be observed that the output direction of the THz wave primarily depends on the discontinuous gradient phase of the designed supercell. When *d*Φ(*x*)*/dx* equals zero, the designed supercell only generates normal reflection for the normal incident THz wave. For practical simulation, the designed supercell is assumed to be in free space (*n_i_* = *n_r_* = 1), and the source is the normal incident LCP wave (*θ_i_* = 0). Thus, the reflection angle can be obtained by simplifying Equation (2) as follow:(3)$$\theta_{r}=\sin^{-1}\left(\frac{\lambda g_{0}}{2\pi p}\right)$$
where *g*_0_ and *p* are the phase gradient along the *x* direction and the periodicity of the unit cell, respectively. The theoretical value of the reflective deflection angle can be calculated by easily employing the simplified form of the generalized Snell law from Equation (3).

To confirm the continuous tunability of the THz beam deflection, the MS supercell is arranged with periodic boundary conditions in both the *x* and *y*-axis directions during simulation. In our designed MS supercell, the *g*_0_ is 45°, denoting the gradient phase. According to Equation (3), the theoretically calculated reflective deflection angle is about 35.2° and −15.3° at 0.65 THz and 1.37 THz, respectively.

Figure 6 and Figure 7 present the deflected electric field (|*E*|) distributions and corresponding normalized intensity of the orthogonal CP wave of the proposed MS supercell with different conductivity of photoconductive Si (*σ*_si_ = 1 S/m, 5 × 10^3^ S/m, 1 × 10^4^ S/m, 2 × 10^4^ S/m, 4.5 × 10^4^ S/m, 1.5 × 10^5^ S/m) at 0.65 THz and 1.37 THz, respectively. The electric conductivity of photoconductive Si can be adjusted easily by changing the infrared pumping illumination power [49,50,51,52,53]. It is evident that the spatial deflection of the electric field results in the same reflective deflection angle at the same frequency but the corresponding normalized intensity decreases gradually when the electric conductivity of photoconductive Si decreases. As shown in Figure 6a, at 0.65 THz, when the MS with photoconductive Si has electric conductivity *σ*_si_ = 1 S/m without infrared pumping illumination, the very weak electric field is directly reflected along the vertical direction without deflection. This is because the photoconductive Si is in an isolation state in the case without infrared pumping illumination, and the whole MS functioned as a mirror reflector. As illustrated in Figure 6b–f, when *σ*_si_ = 1 S/m, 5 × 10^3^ S/m, 1 × 10^4^ S/m, 2 × 10^4^ S/m, 4.5 × 10^4^ S/m and 1.5 × 10^5^ S/m, the photoconductive Si is gradually changed from the isolation to metal state, the reflective deflected electric field distributions will become more and more obvious, and the deflection angle is always kept at about 35°, very close to the theoretical predication value of 35.2°. As depicted in Figure 6g, the normalized intensity of the deflected orthogonal CP wave beam will gradually increase with the increase of the electric conductivity of photoconductive Si. When *σ*_si_ = 1 S/m without infrared pumping illumination, the normalized intensity of the deflected beam is near zero, which is consistent well with the electric-field distribution. When *σ*_si_ increases from 5 × 10^3^ S/m to 1.5 × 10^5^ S/m, the normalized intensity of the deflected beam is increased from 0.1 to 1. This means that the efficiency of the deflected beam is highly dependent on the variation of the infrared pumping illumination power.

As shown in Figure 7, the similar electric field (|*E*|) distributions and normalized intensity of the deflected orthogonal CP beam can also be observed at the higher frequency of 1.37 THz. As depicted in Figure 7a–f, when *σ*_si_ is changed from 1 S/m to 1.5 × 10^5^ S/m, the strength of the reflective deflected electric-field distributions will increasingly intensify, while the deflection angle remains consistently around −15°, closely approximating the theoretical value of −15.2°. As shown in Figure 7g, with the increase of the electric conductivity of photoconductive Si, the normalized intensity of the deflected orthogonal CP wave beam will gradually increase too. When the *σ*_si_ is raised from 1 S/m to 1.5 × 10^5^ S/m, the normalized intensity of the deflected beam rises from 0.1 to 1, also indicating a strong reliance of the intensity of the deflected beam on changes of infrared pumping illumination power at the higher frequency of 1.37 THz.

To further illustrate the tunable reflection deflection effect of the designed MS, as shown in Figure 8, we present the simulated and theoretically calculated (dash line) reflection amplitude of the deflected orthogonal CP wave beam as functions of frequency and deflection angles for the designed MS with photoconductive Si of *σ*_si_ = 1 S/m, 2 × 10^4^ S/m and 1.5 × 10^5^ S/m at the lower and higher frequency regions, respectively. From Figure 8, it is seen that the ordinary reflection (*θ_r_* = 0°) is practically zero at the lower and higher frequency regions of 0.4–0.9 THz and 1–1.6 THz, and only the deflection CP wave beams are reflected with wide angle ranges of 17°–71° and 13°–23°, respectively, according to the generalized Snell law. Note that the theoretical calculation frequency deflection angles are the same when the designed MS with the photoconductive Si has a different electric conductivity at the same frequency range. It is evident that the frequency-dependent reflection deflection angle calculated using Equation (3) aligns well with the simulation results. From Figure 8a–f, it further confirms that the strength of the reflective deflected electric field will increasingly intensify when the electric conductivity of the photoconductive Si increases at both lower and higher frequency regions.

## 4. Conclusions

In conclusion, we have numerically demonstrated a photo-excited MS based on the hybrid patterned photoconductive Si structures in the THz region, which can independently realize the tunable reflective CP conversion and beam-deflection effect at two different frequencies. Numerical simulations show that the reflective CP conversion for the lower frequency region is controlled by the CDSR structure photoconductive Si of the designed MS, while the one of the higher frequency region is determined by the ESP structure. The electric conductivity of the structurally photoconductive Si can be modulated by changing the external infrared-beam pumping power. When the structurally photoconductive Si is in a metal state, the designed MS can convert the incident CP wave into its orthogonal components at two independent frequency regions. By increasing the electric conductivity of the structural photoconductive Si, the designed MS can realize a reflective CP conversion efficiency from 0 to 96.6% at 0.65 THz, and from 0 to 89.3% at 1.37 THz, respectively, and the corresponding modulation depth is up to 96.6% and 89.3%, respectively. Furthermore, the 2π phase shift also can be achieved by respectively rotating the oriented angle (*α_i_*) of the ESP and CDSR at two independent frequency regions. Finally, an MS supercell is constructed by carefully arranging the ESP and CDSR structures to achieve a reflective deflection effect, and the efficiency is tuned continuously from 0 to 99% at the two independent frequencies. The proposed MS in this study has a remarkable photo-excited response, which indicates its promising potential for deployment in active functional THz wavefront devices, including modulators, switches, and deflectors.

## Figures and Tables

**Figure 1 nanomaterials-13-01846-f001:**
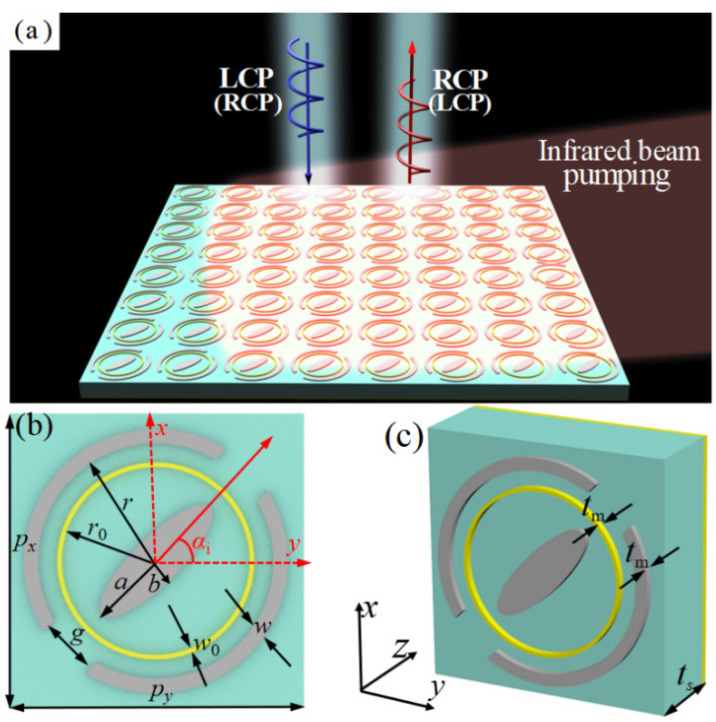
(**a**) Schemes of the proposed photo-excited MS: (**a**) periodic array, (**b**,**c**) front and perspective views of the unit cell.

**Figure 2 nanomaterials-13-01846-f002:**
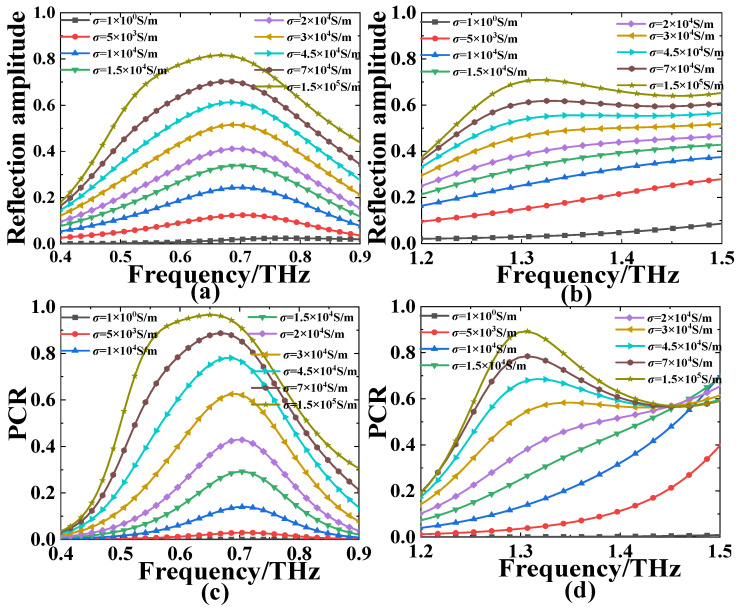
The (**a**,**b**) reflection amplitude of the orthogonal CP wave and (**c**,**d**) the PCR at the (**a**,**c**) lower and (**b**,**d**) higher frequency regions for the proposed MS with different conductivity of photoconductive Si (from *σ*_si_ = 1 S/m to *σ*_si_ = 1.5 × 10^5^ S/m).

**Figure 3 nanomaterials-13-01846-f003:**
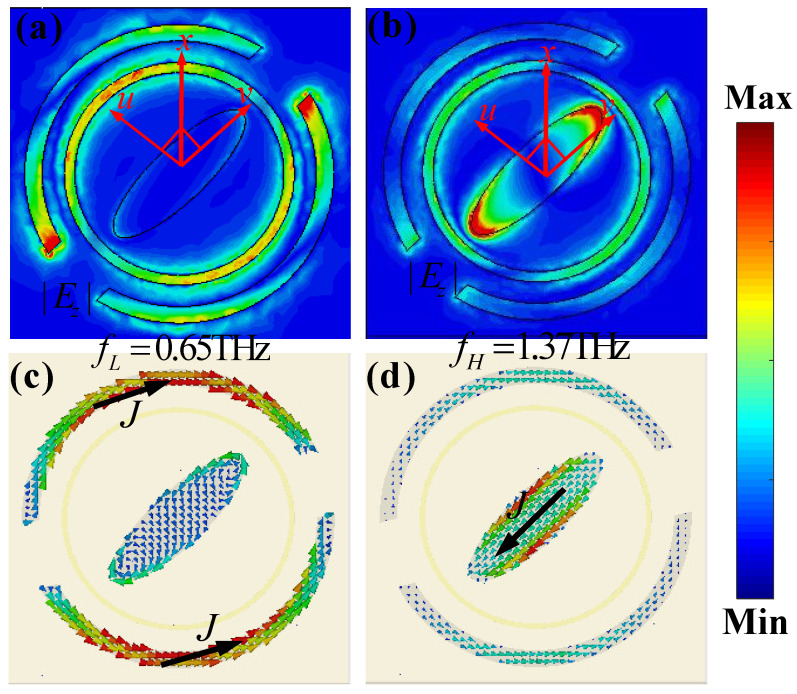
The distributions of the induced (**a**,**b**) electric fields (|*E_z_*|) and (**c**,**d**) surface current of the proposed MS unit cell with photoconductive Si of conductivity σ_si_ = 1.5 × 10^5^ S/m for the normal incident LCP wave at (**a**,**c**) 0.65 THz and (**b**,**d**) 1.37 THz.

**Figure 4 nanomaterials-13-01846-f004:**
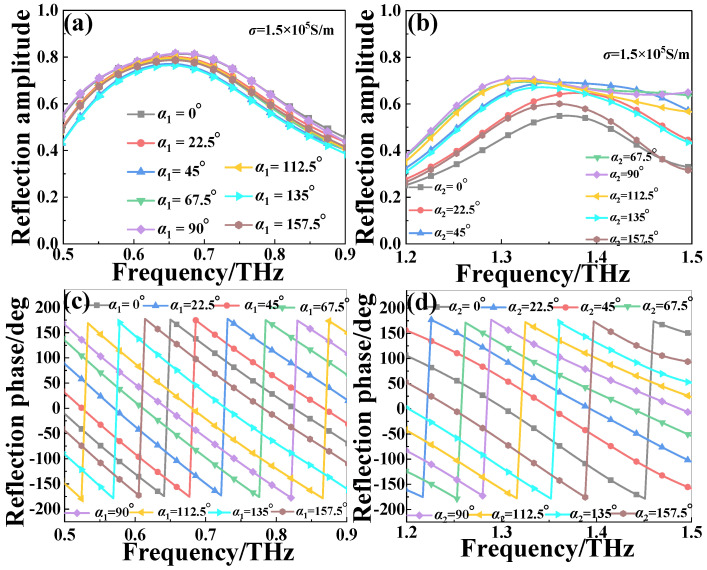
The reflection (**a**,**b**) amplitude and phase (**c**,**d**) of the orthogonal CP wave of the proposed MS structure with different rotation angles (*α*_1_ and *α*_2_) with photoconductive Si of conductivity *σ*_si_ = 1.5 × 10^5^ S/m.

**Figure 5 nanomaterials-13-01846-f005:**
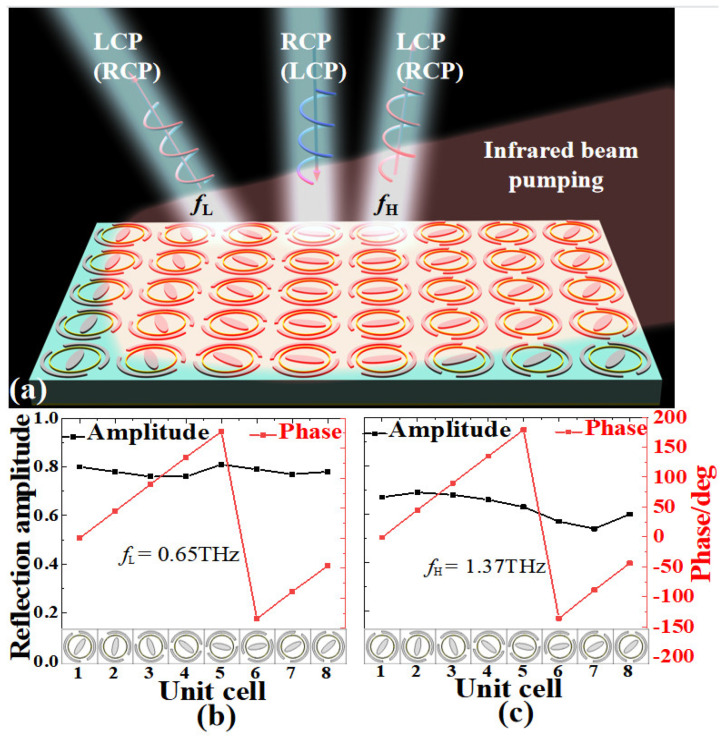
(**a**) The scheme of the designed MS for the tunable THz deflector, the reflection amplitude, and the phase of the orthogonal CP wave of the proposed structure with photoconductive Si of conductivity *σ*_si_ = 1.5 × 10^5^ S/m at (**b**) *f*_L_ = 0.65 THz and (**c**) *f*_H_ = 1.37 THz.

**Figure 6 nanomaterials-13-01846-f006:**
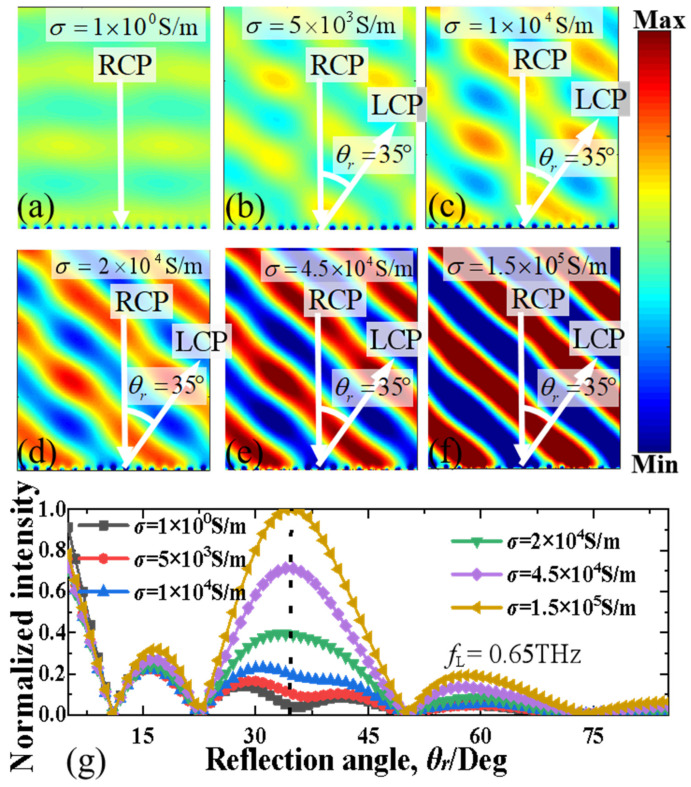
(**a**–**f**) The deflected electric field (|*E*|) distributions and (**g**) normalized intensity of the deflected orthogonal CP beam for the proposed MS supercell with different conductivity of photoconductive Si (from σ_si_ = 1 S/m to σ_si_ = 1.5 × 10^5^ S/m) at the lower frequency of 0.65 THz.

**Figure 7 nanomaterials-13-01846-f007:**
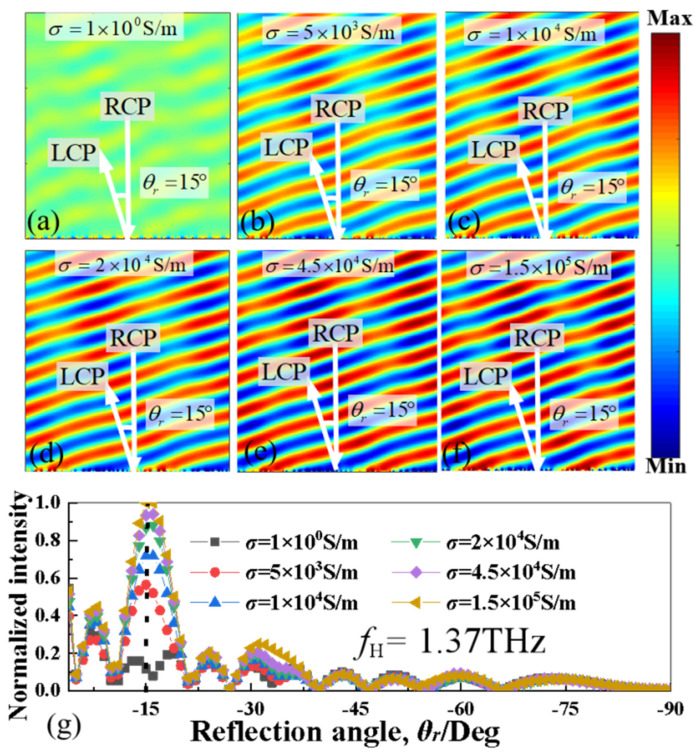
(**a**–**f**) The electric field (|*E*|) distributions and (**g**) normalized intensity of the deflected orthogonal CP beam for the proposed MS supercell with different conductivity of photoconductive Si (from σ_si_ = 1 S/m to σ_si_ = 1.5 × 10^5^ S/m) at the higher frequency of 1.37 THz.

**Figure 8 nanomaterials-13-01846-f008:**
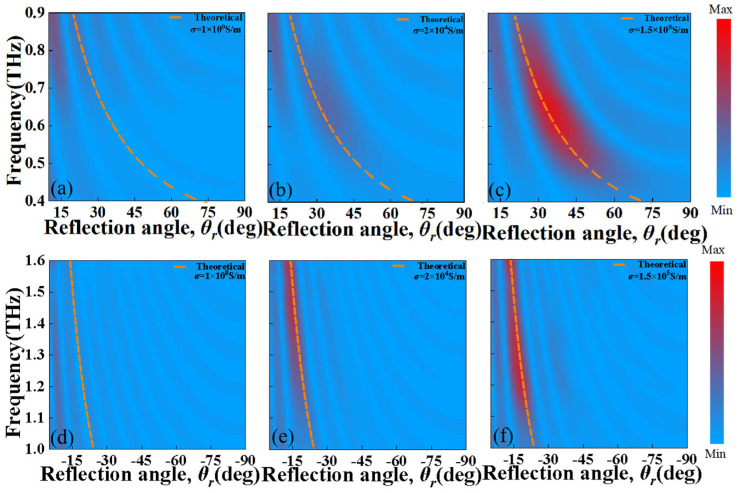
The simulated and theoretical calculated (dash line) reflection amplitude of the deflected orthogonal CP beam as functions of frequency and deflection angles for the designed MS with photoconductive Si of (**a**,**d**) σ_si_= 1 S/m, (**b**,**e**) σ_si_ = 2 × 10^4^ S/m and (**c**,**f**) σ_si_ = 1.5 × 10^5^ S/m at (**a**–**c**) the lower and (**d**–**f**) the higher frequency regions.

## Data Availability

Not applicable.

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
