# Peer review of "Photo-Excited Metasurface for Tunable Terahertz Reflective Circular Polarization Conversion and Anomalous Beam Deflection at Two Frequencies Independently"

_nanomaterials, 2023, doi:10.3390/nano13121846_

Round 1

Reviewer 1 Report

The authors propose a design for tunable MS at THz for anomalous beam deflection and polarization conversion. The paper has serious flows listed below and to my opinion is not suitable for publication, especially on a high-level Journal as Nanomaterials.

1. The paper shows simple numerical simulations of the proposed structure, but they do not provide details of the parameters adopted in the simulation (mesh, geometry, solver, etc). Furthermore, at line 165, they wrote about a generic "simulation optimization" providing the final geometrical parameter, but they do not explain what kind of optimization has been performed nor what is the objective function maximized during the optimization process.

2. Although in the conclusions the authors state that "we have numerically and theoretically demonstrated...", the theoretical part is completely missing since they do not provide any physical insight into the illustrated results. For example, at the beginning they wrote about a "likely Fabry-Perot resonance cavity", then they show the Ez (why not Ex and Ey?) map without analyzing the kind of resonances (if any).

3. The authors introduce two angles \alpha_i, i=1,2, but \alpha_1 and \alpha_2 are related to each other, so it is not clear why they do so.

4. Last, the authors do not provide a motivation for their work and they do not mention what applications they have in mind for the proposed device. 

The quality of the English language is low.

Author Response

The authors propose a design for tunable MS at THz for anomalous beam deflection and polarization conversion. The paper has serious flows listed below and to my opinion is not suitable for publication, especially on a high-level Journal as Nanomaterials.

Authors reply: We thank the reviewer for this concern. We don’t agree with you that the paper is not suitable for publication on journal of nanomaterials.

1) The paper shows simple numerical simulations of the proposed structure, but they do not provide details of the parameters adopted in the simulation (mesh, geometry, solver, etc). Furthermore, at line 165, they wrote about a generic "simulation optimization" providing the final geometrical parameter, but they do not explain what kind of optimization has been performed nor what is the objective function maximized during the optimization process.

Authors reply: We thank the reviewer for this concern. In our study, a full wave simulation was carried out using the frequency solver by the CST Microwave Studio based on the finite element method (FEM). In simulation, the Floquet ports for the incident THz CP plane waves along the z-axis direction are assigned to the unit-cell of the proposed metasurface (MS) structure. In addition, the distance between the wave ports and the unit-cell structure is 300μm in order to only get the enough propagating information of the reflected THz wave. In addition, the minimized mesh size is set as 0.22 μm, which is much smaller than the unit-cell size and the operating wavelength. During the optimization process, the main objective is to get the maximized magnitude of the reflected orthogonal circular polarization (CP) wave at two distinct THz frequency ranges when the designed MS is under the upper infrared pumping power (σsi = 1.5×105 S/m). Note that in the simulation optimization, when the designed MS is under the upper infrared pumping power (σsi = 1.5×105 S/m), the maximized magnitude of the reflected orthogonal CP wave at two distinct THz frequency ranges can be obtained easily just by adjusing the thickness of the middle dielectric layer, the size of the Si ellipse shaped patch (ESP) and circular double split ring (CDSR) and metal circular ring (CR) structure. After simulation optimization, and the final geometrical parameters of the unit-cell are given as: px=py=100μm, a=26μm, b=8μm, r=40μm, w=5μm, g=20μm, tm=3μm, r0=33.5μm, w0=1.5μm, ts=35μm.

2) Although in the conclusions the authors state that "we have numerically and theoretically demonstrated...", the theoretical part is completely missing since they do not provide any physical insight into the illustrated results. For example, at the beginning they wrote about a "likely Fabry-Perot resonance cavity", then they show the Ez (why not Ex and Ey?) map without analyzing the kind of resonances (if any).

Authors reply: We thank the reviewer for this concern. In this paper, we mainly numerically demonstrated a photo-excited metasurface based on hybrid patterned photoconductive silicon (Si) structures for tunable reflective circular polarization (CP) conversion and beam deflection effect at two different THz frequencies. Thus, we agree with you that the theoretical part is completely missing in this work since the relative physics mechanism and physical insight of the reflection-mode polarization conversion and wavefront manipulation have been illustrated detailly and clearly in previous works [S1-S3]. For example, the top and bottom layers of the proposed metasurface structure can form a Fabry-Pérot–like resonance cavity based on interference theory, leading to destructive interference between the co- and cross-polarizations in multi-reflection. It can be expected that the consequent destructive interference of polarization couplings in the multi-reflection process will enhance overall reflected waves with cross-polarizations and reduce the ones with co-polarizations. The previous researches indicate that the interference theory based on Fabry-Perot-like resonance cavity provides a good explanation of broadband reflective polarization conversion[S1]. In addition, to get an intuitive understanding of the operation principle of the designed metasurface structure, as shown in the inset of Figs. 3(a,b), the electric fields of both the incident and the reflected CP wave are decomposed into two orthogonal components, i.e. u-component and v-component, respectively. The eigen-mode resonances on the Si ellipse shaped patch (ESP) and circular double split ring (CDSR) structures can be excited with electric components along u- and v-axis directions, respectively. The excited dipole of the Si ESP and CDSR partly couples to the dipole mode on the interface of the top layer and air, leading to cross-polarization conversion after reflection. The Si ESP and CDSR structures have similar electric responses at different resonance frequencies since it can be considered as evolved from cut-wire structure resonators. Obviously, the z-component of electric field (|Ez|) distributions on the Si ESP and CDSR structures reveal effective excitation of the the surface plasmon polaritons (SPPs) at resonances [S2]. Note that, generally, only the |Ez| distributions can effectively reveal the eigen-mode resonance of the metasurface structure since the |Ez| component correlates with the charge distribution.

[S1]Zhao, J.C.; Cheng, Y.Z.; Cheng, Z.Z. Design of a photo-excited switchable broadband reflective linear polarization conversion metasurface for terahertz waves. IEEE Photonics J. 2018, 10, 4600210.

[S2]Cheng, Y.Z.; Yang, D.R.; Li, X.C. Broadband reflective dual-functional polarization convertor based on all-metal metasurface in visible region. Physical B: Condensed Matter. 2022, 640, 414047.

[S3] Fan, J.P.; Cheng, Y.Z.; He, B. High-Efficiency ultrathin terahertz geometric metasurface for full-space wavefront manipulation at two frequencies. Phys. D: Appl. Phys. 2021, 54, 115101.

3) The authors introduce two angles \alpha_i, i=1,2, but \alpha_1 and \alpha_2 are related to each other, so it is not clear why they do so.

Authors reply: We thank the reviewer for this concern. The azimuth rotation angles of the Si ellipse shaped patch (ESP) and circular double split ring (CDSR) structures of the metasurface unit-cell respect to the the x(y)-axis direction are labled as α1 and α2, respectively. In our design, the two azimuth rotation angles α1 and α2 are independent of each other. The phase shift of reflected orthogonal circular polarization (CP) wave at the lower and higher frequencies can be modulated independently by adjusting the α1 and α2, respectively.

4) Last, the authors do not provide a motivation for their work and they do not mention what applications they have in mind for the proposed device. 

Authors reply: We thank the reviewer for this concern. Metasurfaces (MSs) are currently the subject of extensive research due to their remarkable capacity to flexibly manipulate the polarization and wavefront of electromagnetic (EM)/light waves, which presents a host of significant advantages, such as ultra-thin structures, low losses, and ease fabrication. However, conventional MSs typically possess fixed functionalities that cannot be altered once they are designed and fabricated, thereby constraining their potential applications, especially in terahertz (THz) region. Hence, in this article, we aim to present and demonstrate a new sceheme to realize a optically tunable devices for THz circular polarization (CP) conversion and beam deflection effect. The MS suggested in this study has a remarkable photo-excited response, which indicates its promising potential application for deployment in dynamic functional THz wavefront devices, including modulators, switches, and deflectors. In addition, our proposed photo-excited MSs for the tunable reflective circular polarization converter and beam deflector are highly useful devices in the field of THz technology. They enable the manipulation of THz radiation by controlling its polarization and direction. This has several practical applications, including THz communication systems, imaging and spectroscopy, security screening, and material characterization. By providing tunable control over THz beams, these devices enhance the efficiency, quality, and accuracy of THz-based applications, enabling advancements in THz technology for various industries and scientific research.

Reviewer 2 Report

In paper entitled " Photo-excited metasurface for tunable terahertz reflective circular polarization conversion and anomalous beam deflection at two frequencies independently”, Zhixiang Xu et al demonstrate a photo-excited THz tunable metasurface, which enables circular polarization conversion in reflective configuration and beam deflection at two independent THz frequency ranges. The tunability is based on the optical control of the silicon conductivity, which can be adjusted by changing the optical pump power.

 I think the paper deserves publishing and may be of interest for the metamaterials community. However, I suggest authors to address two points. First, they should refer papers published by the Univ Tokyo group (M. Gonokami et al) on the light-controlled chirality of the silicon structures. This group has demonstrated light-controlled THz circular dichroism a long time ago. Second, the control reflectivity in two frequency ranges simultaneously is an interesting result. However, it looks like that it was obtained from numerical analysis only. Is it possible to find out  relationship between positions of these frequency regions and the silicon conductivity? Presenting simple arguments that may give an idea on the at what extent the conductivity should be changed would strongly improve the paper impact. Another question is how crucial the proposed – rather complicated – design of the unit cell to control of the reflectivity in two regions simultaneously? In the conclusion (page 11), authors state that “…we have numerically and theoretically demonstrated a photo-excited MS…”, however my feeling is that theoretical description has not been developed.

Author Response

In paper entitled " Photo-excited metasurface for tunable terahertz reflective circular polarization conversion and anomalous beam deflection at two frequencies independently”, Zhixiang Xu et al demonstrate a photo-excited THz tunable metasurface, which enables circular polarization conversion in reflective configuration and beam deflection at two independent THz frequency ranges. The tunability is based on the optical control of the silicon conductivity, which can be adjusted by changing the optical pump power.

I think the paper deserves publishing and may be of interest for the metamaterials community. However, I suggest authors to address two points.

Authors reply: We thank you very much for the encouraged comments and some constructive suggestions to this manuscript, we will revise it carefully.

1.First, they should refer papers published by the Univ Tokyo group (M. Gonokami et al) on the light-controlled chirality of the silicon structures. This group has demonstrated light-controlled THz circular dichroism a long time ago.

Authors reply: We thank the reviewer for this concern. We have studied the some works [39,40,58,59] from the Univ Tokyo group and added to the references of our manuscript.

[39]Kan, T.; Isozaki, A.; Kanda, N.; Nemoto, N.; Konishi, K.; Takahashi, H.; Kuwata-Gonokami, M.; Matsumoto, K.; Shimoyama, I. Enantiomeric switching of chiral metamaterial for terahertz polarization modulation employing vertically deformable MEMS spirals. Nature communications. 2015, 6, 8422.

[40]Konishi, K.; Kan, T.; Kuwata-Gonokami, M. Tunable and nonlinear metamaterials for controlling circular polarization. Journal of Applied Physics. 2020, 127, 230902.

[58]Kanda, N.; Konishi, K.; Kuwata-Gonokami, M. All-photoinduced terahertz optical activity. Opt. Lett. 2014, 39, 3274-7.

[59]Kanda, N.; Konishi, K.; Kuwata-Gonokami, M. Dynamics of photo-induced terahertz optical activity in metal chiral gratings. Opt. Lett. 2012, 37, 3510-2.

2.Second, the control reflectivity in two frequency ranges simultaneously is an interesting result. However, it looks like that it was obtained from numerical analysis only. Is it possible to find out  relationship between positions of these frequency regions and the silicon conductivity?

Authors reply: We thank the reviewer for this concern. We agree with you that the control reflectivity in two frequency ranges simultaneously is an interesting result obtained from numerical analysis only. Note that there is no relationship between positions of these frequency regions and the silicon conductivity. The positions of these frequency regions are only determined by the geometric parameters of the metasurface unit-cell structure and the permittity of the middle dielectric layer.

3.Presenting simple arguments that may give an idea on the at what extent the conductivity should be changed would strongly improve the paper impact.

Authors reply: We thank the reviewer for this concern. We agree with you that presenting simple arguments that may give an idea on the at what extent the conductivity should be changed would strongly improve the paper impact. The electric conductivity of the photoconductive Si is usually proportional to the pumping power of the incident infrared light [S1,S2]. For example, the conductivity of the photoconductive Si is only about 1 S/m without infrared pumping (power is zero), while the one is about 1.5×105 S/m under the upper limiting infrared pumping (central wavelength of 800 nm and power is about 405 mJ/cm2) [S1-S3]. This means that the maximal conductivity of the photoconductive Si is up to 1.5×105 S/m when using the upper limiting infrared pumping of 405 mJ/cm2. In addition, the conductivity of the photoconductive Si can be changed from 1.5×105 S/m to 1 S/m when decreases the infrared pumping power.

[S1]Seren, H.R.; Keiser, G.R.; Cao, L.; Zhang, J.; Strikwerda, A.C.; Fan, K.; Metcalfe, G.D.; Wraback, M.; Zhang, X.; Averitt, R.D.  Optically modulated multiband terahertz perfect absorber. Adv. Opt. Mater. 2014, 2, 1221-1226.

[S2]Cheng, Y.Z.; Gong, R.Z.; Zhao, J.C. A photoexcited switchable perfect metamaterial absorber/reflector with polarization-independent and wide-angle for terahertz waves. Optical Materials. 2016, 62, 28-33.

[S3]Zhao, X.G.; Wang, Y.; Schalch, J.; Duan, G.W.; Cremin, K.; Zhang, J.D.; Chen, C.X.; Averitt, R.D.; Zhang, X. Optically Modulated Ultra-Broadband All Silicon Metamaterial Terahertz Absorbers. ACS Photonics. 2019, 6, 830−837.

4.Another question is how crucial the proposed – rather complicated – design of the unit cell to control of the reflectivity in two regions simultaneously?

Authors reply: We thank the reviewer for this concern. In our design, the unit-cell of the proposed photo-excited metasurface (MS) is composed of hybrid structures of Si ellipse shaped patch (ESP) and circular double split ring (CDSR) and metal circular ring (CR) structure, a middle dielectric substrate and a metal ground plane. Note that the two distinct resonators (ESP and CDSR structure Si) are crucial to responsible for manipulating the THz wave at the lower and higher THz frequencies independently. In addition, the metal CR structure is also important to reduce cross talk and cross-coupling effect between the ESP and CDSR structures. By constantly changing the radius of the metal CR, we ultimately controlled the impact to a low degree. Thus, it can be expected that the designed MS can independently control the reflective beams at two different THz frequency regions.

5.In the conclusion (page 11), authors state that “…we have numerically and theoretically demonstrated a photo-excited MS…”, however my feeling is that theoretical description has not been developed.

Authors reply: We thank the reviewer for this concern. We agree with you that the theoretical description has not been developed. So, we have revised the above sentence “…we have numerically and theoretically demonstrated a photo-excited MS…” to “…we have numerically demonstrated a photo-excited MS…” 

Author Response

The paper is written about active metamaterial structures, and the authors claim that “tuneable reflective circular polarization conversion is achievable via beam deflection effect at two frequencies independently”. The topic is interesting and important for applications. Before publication it would be important to clarify the novelty of the structure and to compare the achievable efficiency to those of the previous similar polarization converters and deflectors. The paper is suitable for publication provided the following questions are addressed:

Authors reply: We thank you very much for the encouraged comments and some constructive suggestions to this manuscript, we will revise it carefully. The novelty of the proposed structure lies in its use of a photo-excited metasurface (MS) based on hybrid patterned photoconductive silicon (Si) structures in the terahertz (THz) region. This MS design enables tunable reflective circular polarization (CP) conversion and beam deflection independently at two frequencies. The unit-cell of the MS incorporates metal circular-ring (CR), Si ellipse-shaped-patch (ESP) and circular-double-split-ring (CDSR) structures, along with dielectric substrate and metal ground plane. By adjusting the infrared beam pumping power, the electric conductivity of the Si ESP and CDSR components can be modified. This enables achieving reflective CP conversion efficiency ranging from 0% to 96.6% at a lower frequency of 0.65 THz and from 0% to 89.3% at a higher frequency of 1.37 THz. The proposed MS also achieves high modulation depths of 96.6% and 89.3% at the respective frequencies. Additionally, by rotating the oriented angle (αi) of the Si ESP and CDSR structures, a 2π phase shift can be achieved at the lower and higher frequencies. For beam deflection, a MS supercell is constructed, allowing dynamic tuning of the efficiency from 0% to 99% at the two independent frequencies. With its excellent photo-excited response, this MS holds potential for active functional THz wavefront devices such as modulators, switches, and deflectors.

1) The first sentence in the abstract could be moved into the Introduction section.

Authors reply: We thank the reviewer for this concern, we have moved the first sentence in the abstract into the introduction section.

2) The authors state that “reflective circular polarization conversion is achievable “via beam deflection effect”, but it would be more reasonable to insert “and”, considering that these are coexistent capabilities.

Authors reply: We thank the reviewer for this concern. We agree with you that using the “and” to replace the “via” to point out that they are coexistent.

3) There are different numbers quantifying the “CP conversion efficiency”, and “modulation depth”, it would be important to clarify the meaning of the latter, considering its larger value compared to the CP conversion efficiency interval.

Authors reply: We thank the reviewer for this concern. We agree with you that it would be important to clarify the meaning of “modulation depth” in our manuscript.

To further illustrate the tunability of the proposed MS structure for the CP conversion, the modulation depth of the CP conversion is defined as d=PCRmax-PCRmin, where the PCRmax and PCRmin are the CP conversion efficiency when the proposed MS structure is under the upper limiting infrared pumping power (σsi = 1.5×105 S/m) and without infrared pumping power (σsi = 1 S/m). Thus, the corresponding modulation depth of the CP conversion of the proposed structure is near 96.6% and 89.3% at the lower and higher frequencies, respectively.

  • “Dynamic functional metamaterials” are usually nominated as “active metamaterials,” and in the present case the activation is of photonic origin, explicit expression of this would ensure the correct classification of the device. However, the refereed conductivity modification is very large, and similar values appear mainly in the papers written by the same authors. An experimental work about this dynamic conductivity modification interval should be also cited.

Authors reply: We thank the reviewer for this concern. We agree with that “Dynamic functional metamaterials” are usually nominated as “active metamaterials,” and in the present case the activation is of photonic origin, explicit expression of this would ensure the correct classification of the device. So, we have revised them in our manuscript. In addition, we also have cited some experimental works [49,55] work about this dynamic conductivity modification interval in our manuscript.

[49]Seren, H.R.; Keiser, G.R.; Cao, L.; Zhang, J.; Strikwerda, A.C.; Fan, K.; Metcalfe, G.D.; Wraback, M.; Zhang, X.; Averitt, R.D.  Optically modulated multiband terahertz perfect absorber. Adv. Opt. Mater. 2014, 2, 1221-1226.

[55]Zhao, X.G.; Wang, Y.; Schalch, J.; Duan, G.W.; Cremin, K.; Zhang, J.D.; Chen, C.X.; Averitt, R.D.; Zhang, X. Optically Modulated Ultra-Broadband All Silicon Metamaterial Terahertz Absorbers. ACS Photonics. 2019, 6, 830−837.

5) There are a lot-of versions of similar converters and deflectors in the literature, those proposed in the present paper seem to be unrealistic or extremely hard/expensive to fabricate. The experimental feasibility and the advantages compared to similar, but completely metal structures, should be detailed to justify their usefullnes.

Authors reply: We thank the reviewer for this concern. We agree with you that there are a lot-of versions of similar converters and deflectors in the literatures, but we don’t think that those proposed in the present paper seem to be unrealistic or extremely hard/expensive to fabricate. Significantly different from previous designs, the unit-cell of the proposed photo-excited MS is composed of hybrid structures of Si ellipse shaped patch (ESP) and circular double split ring (CDSR) and metal circular ring (CR) structure, a middle dielectric substrate and a metal ground plane. The proposed photo-excited MS can realize the tunable reflective circular polarization (CP) conversion and beam deflection effect at two frequencies independently by changing the external infrared pumping power. Owing to the limitation of the experimental conditions in our lab, we just provided the numerical simulation results in this manuscript. It is assure that the fabrication of such a simple device in the terahertz (THz)  regime is not complex, in the later, we will consider cooperating with other laboratories to start THz frequency domain experimental work. We can use photolithography, electron-beam metal deposition, and lift-off to fabricate the metallic structures, and spin coating and thermal curing can be used to apply the polyimide layers [S1-S5]. In addition, our proposed photo-excited MSs for the tunable reflective circular polarization converter and beam deflector are highly useful devices in the field of THz technology. They enable the manipulation of THz radiation by controlling its polarization and direction. This has several practical applications, including THz communication systems, imaging and spectroscopy, security screening, and material characterization. By providing tunable control over THz beams, these devices enhance the efficiency, quality, and accuracy of THz-based applications, enabling advancements in THz technology for various industries and scientific research.

[S1]Chen H T, Padilla W J, Zide J M O, Gossard A C, Taylor A J and Averitt R D 2006 Active terahertz metamaterial devices, Nature 444 597–600.

[S2]H.T. Chen, J.F. O’Hara, A.K. Azad, A.J. Taylor, R.D. Averitt, D.B. Shrekenhamer, W.J. Padilla, Experimental demonstration of frequency-agile terahertz metamaterials, Nature Photon. 2 (2008) 295.

[S3] T. Niu, W. Withayachumnankul, B. S.-Y. Ung, H. Menekse, M. Bhaskaran, S. Sriram, and C. Fumeaux, Opt. Express 2013, 21(3), 2875-2889

[S4] N. K. Grady, J. E. Heyes, D. R. Chowdhury, Y. Zeng, M. T. Reiten, A. K. Azad, A. J. Taylor, D. A. R. Dalvit, H. T. Chen, Science 2013, 340(6138), 1304-1307.

[S5] R. H. Fan, Y. Zhou, X. P. Ren, R. W. Peng, S. C. Jiang, D. H. Xu, X. Xiong, X. R. Huang, and M. Wang, Adv. Mater. 2015, 27(7), 1201-1206.

6) The authors claim that the limit in device efficiency is caused by the “strong-coupling” of the elements, but there is no system specific coupling constant defined that should outperform the involved resonances related characteristics threshold. Is there a strong-coupling or a crosscoupling, considering the spectral and spatial symmetry properties of the modes?

Authors reply: We thank the reviewer for this concern. We agree with you that there is no system specific coupling constant defined that should outperform the involved resonances related characteristics threshold. It is assure that there is a cross-coupling between the Si ellipse shaped patch (ESP) and circular double split ring (CDSR) of the proposed metasurface structure. To reduce cross talk and cross-coupling effect between the ESP and CDSR structures, the metal CR structure is added and used in our design. To further illustrate this cross-coupling, considering the spectral and spatial symmetry properties of the modes, as shown in FS1, we present the reflection spectra of the orgthogonal circular polarization (CP) wave and PCR of the MS unit-cell based on the ESP, ESP+CSDP, CDSP and the proposed structure. As shown in FS1.(b,c) although the MS based on the only CDSP and the combination of the ESP and CDSP structures can achieve the reflective CP conversion in a broadband frequency range, and the efficiency is relative low at the lower and the higher frequencies, which can’t adjust independently the CP conversion at two different THz frequencies due to the cross-coupling effect between the ESP and CDSR structures. In addition, the MS based on the only ESP structure achieves a relatvie low-efficient CP converion at the higher frequency region. When introducing the metal CR structure into our design, the proposed MS can achieve the high-efficient reflective CP conversion at the lower and the higher frequencies independently and simultaneously.

FS1. (a) The front views of the MS unit-cell based on the ESP, ESP+CSDP, CDSP and the proposed structure, (b,c) the simulated reflection amplitude of the orgthogonal CP wave and PCR.

7) The Ez component correlates with the charge distribution, however both are time-dependent, therefore |Ez| and the representative charge distribution would be more unambiguous to identify the dominant modes

Authors reply: We thank the reviewer for this concern. We agree with you that the Ez component correlates with the charge distribution, and |Ez| and the representative charge distribution would be more unambiguous to identify the dominant modes. So, we have changed the Ez to |Ez| distributions in Figs.3(a,b), please see the revised version of our manuscript.

Round 2

Reviewer 1 Report

All the issues I raised have been fully addressed and the quality of the paper increased a lot by adding important details and information.

Minor corrections on English Language is advised; proofreading by a native speaker might help.